# Expanding Role of Ubiquitin in Translational Control

**DOI:** 10.3390/ijms21031151

**Published:** 2020-02-09

**Authors:** Shannon E. Dougherty, Austin O. Maduka, Toshifumi Inada, Gustavo M. Silva

**Affiliations:** 1Department of Biology, Duke University, Durham, NC 27708-0338, USA; shannon.dougherty@duke.edu (S.E.D.); austin.maduka@duke.edu (A.O.M.); 2Graduate School of Pharmaceutical Sciences, Tohoku University, Sendai 980-8578, Japan; toshifumi.inada.a3@tohoku.ac.jp

**Keywords:** ubiquitin, ribosome, translation regulation, oxidative stress, quality control, ribosomal protein, degradation

## Abstract

The eukaryotic proteome has to be precisely regulated at multiple levels of gene expression, from transcription, translation, and degradation of RNA and protein to adjust to several cellular conditions. Particularly at the translational level, regulation is controlled by a variety of RNA binding proteins, translation and associated factors, numerous enzymes, and by post-translational modifications (PTM). Ubiquitination, a prominent PTM discovered as the signal for protein degradation, has newly emerged as a modulator of protein synthesis by controlling several processes in translation. Advances in proteomics and cryo-electron microscopy have identified ubiquitin modifications of several ribosomal proteins and provided numerous insights on how this modification affects ribosome structure and function. The variety of pathways and functions of translation controlled by ubiquitin are determined by the various enzymes involved in ubiquitin conjugation and removal, by the ubiquitin chain type used, by the target sites of ubiquitination, and by the physiologic signals triggering its accumulation. Current research is now elucidating multiple ubiquitin-mediated mechanisms of translational control, including ribosome biogenesis, ribosome degradation, ribosome-associated protein quality control (RQC), and redox control of translation by ubiquitin (RTU). This review discusses the central role of ubiquitin in modulating the dynamism of the cellular proteome and explores the molecular aspects responsible for the expanding puzzle of ubiquitin signals and functions in translation.

## 1. Introduction

Ribosomes are one of the most abundant molecular machines in a cell and perform an essential role in gene expression. Although translation regulation is instrumental for cellular physiology, thus far, much of the study on cellular gene expression has been conducted at the level of transcription [1,2,3,4,5,6]. Due to advances in next generation sequencing, RNA-seq has become an efficient way to measure the transcriptome and is commonly used as an estimation for protein levels [7]. Despite its widespread implementation, recent studies suggest that only partial correlation exists between the transcriptome and proteome, ranging from a 0.4–0.7 correlation coefficient in yeast [8,9,10,11]. This range suggests that significant regulation occurs post-transcriptionally, including at the level of translation [12,13,14,15,16]. This is supported by the increased accuracy of newly developed ribosome profiling techniques such as Ribo-seq, which show that ribosome occupancy has a higher correlation range with protein abundance than mRNA levels [11,17,18]. As protein production is an energetically costly process [19] and is responsible for shaping proteome dynamics, regulation at the translational level is essential to ensure proper cellular function and health. Translation can be regulated at many different steps, including initiation, elongation, and termination, with each being fundamental processes conserved across all organisms [20,21,22]. These steps require the proper binding of both RNA and protein factors, and regulation is often facilitated by the alteration of these binding patterns. As ribosomes are highly abundant and responsible for all protein synthesis within the cell, even minor changes to binding patterns could have a large impact on global protein production and cellular health, depending on which steps of translation are altered. Mutations and dysregulation of translational control can lead to a variety of diseases such as cancer, neurological disorders, bone marrow dysfunction and immunodeficiency, among others [23]. As translation is a core process in maintaining cellular function and health, dysfunctional regulation can have devastating results for the cell.

Translational control does not occur in a universal manner across a single cell, but varies based on ribosomal subpopulations, with functions specific to cellular localization, transcript targeting, and signaling pathways [24,25]. These subpopulations are distinguishable by RNA and ribosomal protein composition [26,27], binding factors [28], intracellular localization [29,30], and post-translational modifications (PTM) [31]. These subpopulations have a differential capacity to bind and interact with both mRNA and protein factors required for active translation, and are likely provide distinct occurrences of regulation throughout the cell [24,27]. The addition of post-translational modifications to ribosomal proteins is a fast, and in many instances, reversible way to create subpopulations after ribosome maturation and to allocate ribosomes to perform distinct functions within the cell, such as localizing translation of specific transcripts [24]. One of the most prominent ribosomal PTM is ubiquitin, a highly conserved 76 amino acid protein, which is enzymatically conjugated to a target protein. Although initially characterized as the main marker for protein degradation [32,33,34], a growing body of literature has shown ubiquitin’s ability to regulate multiple processes beyond degradation [35]. Ubiquitin dysfunction has been implicated in several diseases, including neurodegenerative diseases [36,37] as well as several types of cancer [38,39,40], and it has been shown that ubiquitin modifies a large fraction of the human proteome, with over 60,000 sites on more than 9000 distinct proteins identified in mammalian cells [41,42,43]. From this abundance of sites, it is unsurprising that ubiquitin modifies several ribosomal proteins to regulate ribosomal function and abundance, both of which are essential for maintaining cell homeostasis. Recently, ubiquitin was found to play an essential regulatory role in several ribosome processes, including degradation of ribosomal subunits, quality control of arrested peptides and faulty mRNA, and modulation of translation in response to oxidative stress. The growing study of ubiquitin-mediated translational control provides new insights into the mechanisms of maintenance of cellular health and may result in a greater understanding of diseases associated with both the ubiquitin and the translation system. In this review, we will elaborate on these expanding roles of ubiquitin in translational control and discuss the impact of ribosomal ubiquitination in defining the proteome and maintaining cellular health.

## 2. Ubiquitin Specificity

Ubiquitin is a protein modifier that plays an evolutionarily conserved role in regulating function and protein fate in eukaryotic cells [44]. From its discovery, ubiquitin has been characterized as a mark for degradation of proteins [32,33,34]. Beyond targeting proteins for degradation, ubiquitin can control various cellular processes by inducing structural changes, altering protein localization, and regulating protein-protein interactions [45,46,47,48]. These different functions have been observed in virtually all signaling pathways, such as DNA repair [47], endocytosis [45], kinase regulation [46], transcriptional and translational control [48,49,50,51,52], in addition to larger multicellular processes like inflammation [53] and immune system signaling [54]. In this section, we will discuss how the arrangements of ubiquitin chains and different ubiquitin enzymes define pathway specificity and regulate a slew of cellular processes from protein synthesis to degradation.

### 2.1. Ubiquitin Linkages

Ubiquitin’s function is largely determined by the enzyme-dependent arrangements of ubiquitin monomers into chains of various lengths and linkages. In the conjugation reaction, the C-terminus of ubiquitin forms an isopeptide bond with the amino group of a lysine sidechain or the N-terminus of a target protein [32]. Once a target protein has its first ubiquitin monomer, a process also known as monoubiquitination, this ubiquitin moiety itself can be ubiquitinated on one of its lysine sidechains, extending the modification into a polyubiquitin chain (Figure 1). As ubiquitin contains seven lysine amino acids and an N-terminal methionine (M1), these residues can link ubiquitin monomers into several structurally different chains on the target protein (Figure 1A) [55]. These different chain types, also known as ubiquitin linkages, determine the signaling function of the modification [56,57,58]. Polyubiquitin linkages are classified into homotypic, consisting of bonds at a single ubiquitin lysine position; heterotypic, in which the bonds occur at several lysine positions; and branched, which requires ubiquitination at more than two sites on a single ubiquitin molecule [59]. This intricate ubiquitin code allows cells to expand the repertoire of signaling functions mediated by a single molecule (ubiquitin), remarkably contributing to the complexity encoded in eukaryotic genomes.

The number of specific functions of ubiquitin are exponentially increased when the combination of polyubiquitin linkages is considered in addition to the functions of monoubiquitin. Monoubiquitin is known to play a role in transcriptional control, DNA repair, metabolism, and apoptosis [60]. Homotypic chains, the more well-studied types of the polyubiquitin chains, are designated by the lysine position that defines the linkage. For example, K48 ubiquitin is the most abundant linkage type and the canonical signal for proteasomal degradation [61]. K48 ubiquitin consists of a ubiquitin chain covalently bonded to the ε-amino group of the lysine in the 48th position of the preceding ubiquitin molecule [62]. In regards to translation, K48 ubiquitin possesses an important function in the degradation of ribosomal units by the proteasome, ensuring proper ribosomal composition, and removal of excess free protein [49]. K63 ubiquitin, another well-studied homotypic linkage is involved in several signaling pathways [63] and regulates multiple translation process independent of the proteasome, including translation quality control and regulation of translation during oxidative stress [64,65]. K63 ubiquitin is also known to induce autophagy, adding another yet less specific path for degradation of dysfunctional ribosomes by the lysosome [49,66]. Homotypic linkages at the other lysine positions are now beginning to be understood [67]. For example, K11 homotypic chains are involved in the degradation of anaphase-specific proteins to regulate cell cycle progression [68], and M1 linkages influence many signaling pathways [69]. Heterotypic linkages are also understudied, but K11 and K48 heterotypic linkages have recently been implicated in degradation by the proteasome [70,71]. Ubiquitin function can be further specified by the introduction of additional post-translational modifications such as phosphorylation or acetylation, which can increase instances of ubiquitination and inhibit polyubiquitin formation, respectively [72,73,74]. Polyubiquitin chain length is also believed to modify ubiquitin function, and recent techniques have been developed to address its functional impact [75]. The increasingly complex combinations of ubiquitin linkages, chain length, and additional PTMs accounts for the extreme specificity of ubiquitin, which is essential for fostering correct protein-protein interactions.

The countless combinations and structures of ubiquitin binding domains (UBDs) allow for the specialized functions of specific ubiquitin linkages found in an array of pathways, including several aspects of translational control. Proteins containing a UBD can interact with particular ubiquitin moieties to provide functional specificity, serving as a reader of the ubiquitin signal. There are more than 20 different families of UBDs with varying structural motifs that provide the necessary structural interface for specific binding of ubiquitinated targets [76]. For example, one family of UBD known as ubiquitin associated domains (UBA) recognize specifically K48 polyubiquitin chains in both mammalian and yeast proteins [77,78]. Another UBD, the ubiquitin interacting motif (UIM), binds selectively to K63 polyubiquitin chains, which is required for example in ubiquitin-mediated DNA repair [79]. The combination of different UBDs on a single ubiquitin receptor molecule provides even more possibilities for recognition and binding of specific ubiquitin chains, resulting in ubiquitin linkage-specific signaling and functions. Ubiquitin linkage diversity allows for an array of highly specialized roles in translation via modification of the ribosome, which we will discuss in depth later in this review.

### 2.2. Ubiquitin Enzymatic Cascade

A second source of ubiquitin signal specificity is determined by the plethora of enzymes responsible for recognizing selective targets and catalyzing the ubiquitin conjugation reaction. Formation of ubiquitin chains on a target protein requires a cascade of increasingly specific enzymes, including an activating enzyme (E1), conjugating enzyme (E2), and a ligase (E3) (Figure 1B). E1 activates free ubiquitin in an ATP-dependent reaction in the initial step of the enzyme cascade. E1s are the least specific of the ubiquitinating enzymes, with only one enzyme encoded in yeast (*UBA1*) and two known enzymes encoded in humans (*UBE1* and *UBA6*) [83,84]. Following the activation by an E1, ubiquitin is transferred to a ubiquitin conjugating enzyme (E2). In yeast, Uba1 interacts with all known E2s, while the human homolog UBE1 interacts with all known E2s, excluding USE1, a UBA6-dedicated E2 [84]. Ubiquitin-specific E2s are more diverse than E1s with 11 and over 35 known enzymes encoded in the yeast and human genomes, respectively [85,86]. Despite the increased diversity of E2s, it is the distinct combination of E2-E3 pairs that recognize selective targets and fully define their specificity. Although the E2s are involved in the conjugation of ubiquitin and are responsible for defining linkage specificity, the E3s are the enzymes responsible for recognizing both the protein target and the E2 for proper catalysis of the ubiquitin reaction.

Ubiquitin ligases (E3s) are the largest class in the ubiquitin enzyme cascade with 60–100 putative in yeast and over 600 in humans [85,86,87]. Two major families of E3s exist in eukaryotes, consisting of either a HECT (Homologous to the E6-AP Carboxyl Terminus) or RING (Really Interesting New Gene) finger domain [88]. The HECT domain has a bilobal structure in which the C-terminal lobe interacts with the target protein, and the N-terminal lobe binds to the associated E2 [89]. The HECT ubiquitin ligase directly interacts with the E2 conjugated ubiquitin via a unique thioester bond between a cysteine residue and the ubiquitin C-terminus [89]. RING E3s are more abundant than HECT and aid ubiquitination by providing a scaffold for proper interaction between the E2 and target, allowing for ubiquitination to occur [90]. The RING finger domain consists of a series of cysteine and histidine residue loops, which coordinate the binding of two zinc ions to bind the E2 [91]. The formation of distinct E2-E3 pairs is essential to target proper proteins and form specific ubiquitin chain types [92]. Because of the complexity of ribosome structure and the large number of individual proteins, several E2s and E3s were found that modify the ribosome at distinct residues and at various stages during translation.

The last class in the ubiquitin enzymatic cascade is composed of deubiquitinating enzymes (DUBs) (Figure 1B). DUBs also play an essential role in maintaining global levels of protein ubiquitination by removing ubiquitin from the target or breaking up polyubiquitin chains into free monomers [93]. There are estimated to be around 20 DUBs in yeast and nearly 100 in mammalian cells that fall into five subfamilies based on their proteolytic activity, structural folds, and targets [94,95,96]. Similar to E2s and E3s, DUBs target specific proteins and ubiquitin chain type and act to counterbalance the regulatory processes mediated by ubiquitin [97]. DUB enzymatic specificity is largely dependent on the deubiquitinase subfamily identity. Of the five subfamilies, one (JAMM) is a metalloprotease which requires a catalytically active zinc to cleave ubiquitin linkages, and the remaining four are cysteine proteases, with the USP (ubiquitin specific protease) subfamily being the most prevalent [98]. The cysteine protease families contain a catalytic triad to activate the catalytic cysteine residue and a highly conserved ubiquitin binding site (S1), which orients the ubiquitin moiety to increase the rate of isopeptide bond cleavage [98]. The specificity of DUBs is largely controlled by the number and spacing of S1 sites for ubiquitin linkages as is seen with UBDs, and the presence of specific domains to precisely recognize their protein target [93]. With the extensive combinations of S1 sites and substrate specific domains, DUBs regulate the buildup of ubiquitin in a targeted manner, ensuring the ability to control ubiquitin signaling pathways. The E2–E3 pair and DUB create an antagonistic system that can strictly control the level of ribosomal ubiquitination. As distinct enzymes can produce and remove distinct ubiquitin linkages, they play an important role in determining ubiquitin function in multiple pathways of translational regulation.

### 2.3. Ubiquitin Proteasome System

With a growing understanding of ubiquitin biology, it is clear that protein ubiquitination modulates a variety of signaling pathways in addition to its canonical role in protein degradation. However, in the canonical pathway, ubiquitinated proteins are shuttled to the proteasome for degradation as part of the ubiquitin proteasome system (UPS). The proteasome is a large protease complex responsible for the breakdown of proteins into small peptides and consists of a 20S catalytic core particle (CP) that can be coupled to several different regulatory particles (RP) [99]. One abundant RP is the 19S, a multi-subunit particle that contains a lid and base subcomplex and recognizes ubiquitinated proteins [99]. Polyubiquitin chains are recognized by the yeast protein Rpn10 (regulatory particle non-ATPase 10, or S5a in mammals), which also links the lid and base subcomplexes [100,101]. Multiple DUBs are found to be associated with the 19S that cleave and recycle ubiquitin prior to protein degradation, including Rpn11, Ubp6, and Uch37 [102]. Several E3s have also been found associated to this part of the proteasome, suggesting continuous ubiquitin-mediated regulation of degradation until proteins are unfolded in the base of the 19S [101]. The base of the 19S consists of six AAA-ATPases responsible for the unfolding of proteins and shuttling to the core particle through a narrow channel [103,104]. The catalytic core contains a stack of threonine proteases with active sites facing internally [101]. Both the narrow channel and the internal active sites regulate access to the proteases, preventing degradation of untargeted and unfolded proteins. Although K48 and K11 are abundant and are the main linkages involved in protein degradation, it has been shown that all linkages but K63 accumulate in the presence of proteasome inhibition, suggesting a potential degradative role for all these chains [105]. Degradation can regulate ribosome abundance, clear extra or defective proteins, and control gene expression through co-translational degradation, thus being one of the essential mechanisms of translational control mediated by ubiquitin.

## 3. Role of Ubiquitin in the Control of Ribosome Abundance

In the following sections, we will address the variety of mechanisms by which ubiquitin controls ribosome function and translation. A direct method of translational regulation is to control the abundance of functional ribosomes through synthesis or degradation. Perturbations in ribosome abundance can occur during environmental changes, including cellular response to stresses [106] and limited nutrient availability [107]. As global levels of ribosome abundance are tightly regulated, both ribosome biogenesis and degradation can be regulated by ubiquitin signaling. Ribosome biogenesis itself is an extremely orderly process, involving a cascade of enzymes and several subcellular locations for proper maturation [108]. Together with translocation of ribosomal proteins from the cytosol to the nucleolus, rRNA processing and assembly occur prior to nuclear export and final maturation of translation-competent ribosomes [108]. A number of the processes regulating ribosomal proteins abundance and assemble into the ribosome are facilitated by ubiquitin [109]. Similar for biogenesis, ribosome degradation also involves multiple ubiquitin-mediated mechanisms such as degradation through the UPS [103] and autophagy [49,110]. Rapid degradation of ribosomal proteins is thought to occur through the UPS, while bulk degradation of ribosomal subunits occurs through a selective autophagic pathway called ribophagy, which is also mediated by ubiquitin signaling [111]. Ribosomal RNA (rRNA) are also subject to regulated mechanisms of degradation, specifically when its functionality is lost [112]. In these cases, ubiquitin ligases serve as readers of mutated rRNA, ubiquitinating ribosomal proteins that promotes rRNA degradation [113]. This section will discuss the roles of ubiquitin in the control of ribosome abundance.

### 3.1. Ubiquitin in Ribosome Biogenesis

The ubiquitin moiety has been shown to be integral to the process of ribosome biogenesis. Biogenesis begins in the nucleolus, where immunofluorescence studies have shown that ubiquitin is abundant [114]. From the initial steps, the synthesis of ribosomal proteins is linked to the production of ubiquitin, as multiple ribosomal genes are fused with ubiquitin genes [109]. Ribosomal proteins fused to ubiquitin are seen in all eukaryotes, assisting the assembly of ribosomal proteins into the mature ribosome [109,115]. In humans, two ubiquitin genes encode for polyubiquitin precursor proteins (UBB and UBC) while the other two have ribosomal proteins fused to the C-terminus of ubiquitin (UBA52 and RPS27A). After translation, cleavage of these ubiquitin fusion proteins produces eL40 and eS31, respectively (Figure 2). The mechanism behind how the fused ubiquitin facilitates the assembly of ribosomal proteins or promotes ribosome maturation overall is still unknown; however, one study has suggested that the fused ubiquitin serves as a chaperone to ensure efficient translation of eS31 [116]. Nevertheless, these fusion proteins are essential for ribosome function, as deletion of ubiquitin from the *UBI3* ubiquitin fusion gene in yeast (that produces ubiquitin and eS31) led to defects in maturation of ribosomes [109]. Further supporting the crucial role of these ubiquitin fusion proteins, an siRNA knockdown of the UBA52 transcript in mammals led to a decrease in global protein synthesis [117]. It was further shown that not only the presence, but also the cleavage of ubiquitin from the ribosomal protein is necessary for proper ribosome biogenesis. By inducing mutations in *UBI3* that prevent ubiquitin cleavage of eS31, yeast cells show a decrease in translation initiation, and a delay in pre-rRNA processing [116]. The UPS also has potential involvement in ribosome biogenesis, as proteasomal inhibition by MG-132 impacted overall nucleolar structure and protein dynamics [114]. In vivo studies have also shown that deletion of UBA52 in mice embryos led to death during embryonic development [117], which highlights the importance of this process in cellular health and disease. A refined regulation of ribosome biogenesis serves as the first step of translational control. Although additional research is needed to fully elucidate this pathway, ubiquitin plays an essential role in the progression of the ribosome biogenesis, maturation, and protein production.

### 3.2. Ubiquitin-Mediated Pathways of Ribosomal Protein Degradation

Ribosomal proteins undergo processes of degradation to control the proper stoichiometry necessary to assemble functional ribosomes. Excess ribosomal proteins that have not been incorporated into ribosomes are specifically modified by ubiquitin, which facilitates their degradation through the proteasome [52] (Figure 3). While K48 and K11 linkages are globally considered the main linkages involved in protein degradation [105], the linkage type involved for ribosomal protein degradation remains unconfirmed. In the presence of the proteasome inhibitor bortezomib, it was demonstrated that overexpression of the large subunit component uL24 led to accumulation and aggregation in its polyubiquitinated forms [118]. By screening for 115 UPS-related genes in yeast, the E2 conjugase genes *UBC4*/*UBC5* and E3 ligase gene *TOM1* were found to be involved in the degradation of excessive ribosomal protein [52]. Tom1 contains a HECT-domain, and was previously implicated in cell cycle progression and transcriptional regulation [119]. Depletion of Tom1 in yeast was shown to cause a similar phenotype of ribosomal protein aggregation compared to the use of bortezomib [52]. By using site-directed mutagenesis to disrupt uL24 binding to rRNA and incorporation into mature ribosomes, this group found that Tom1 ubiquitinated residues are usually embedded in the 3D structure of the ribosome [52], providing a rationale for how Tom1 is only involved in the degradation of free ribosomal proteins. Additionally, mapping of all Tom1 ubiquitination sites on the large subunit revealed that 83% of these sites are buried and inaccessible in the mature ribosome [52], preventing their degradation. These findings provide a unique mechanism as to how this E2 conjugase and E3 ligase pair confer specificity to free, excess ribosomal proteins.

In addition to the UPS, autophagic pathways mediated by ubiquitin are also involved in the degradation of ribosomes. Ribosomes undergo autophagy both by random nonselective autophagic engulfment, as well as selective autophagic processes [49]. Ribophagy, a selective autophagic pathway characterized by Kraft and colleagues, is a ubiquitin-mediated mechanism involving the specific delivery of ribosomal subunits to the lysosome for degradation [111] (Figure 3). The pathway is highly sensitive to nutrient availability, suggesting that this translational regulation mechanism serves as a means to minimize energy use [111]. Upon nutrient starvation, the nutrient-sensing megacomplex mTORC1 regulates the flux of substrates for autophagic degradation, including ribosomes [120]. During nitrogen starvation, it was found through genetic screening that Ubp3 and its cofactor Bre5 are both essential for ribophagy to occur [111]. Ubp3 (USP10 in mammals) is a deubiquitinase known to participate in other pathways when complexed with the Ubp3-associating protein Bre5, such as COPII protein deubiquitination [121]. Cdc48 and Ufd3 form a complex with Ubp3, and their deletion also results in defective ribophagy [122]. It was later discovered that the E3 ligase Ltn1 (also involved in ribosome quality control) ubiquitinates the ribosomal protein uL23 at K74, and deletion of Ltn1 rescues the ribophagy-defective phenotype in *UBP3*-deleted yeast cells [123] (Figure 2 and Figure 3). Also, levels of Ltn1 are decreased during starvation conditions [123], providing additional correlations of ribophagy control and nutrient availability. This all suggests that ubiquitination by Ltn1 serves as a signal that prevents ribosome degradation through ribophagy, and the removal by Ubp3 allows for ribophagy to proceed. However, the linkage type involved, as well as how the removal of ubiquitin promotes facilitation into the autophagosome both still remain in question. A model was proposed where the ubiquitin signal blocks the recognition of the ribosome by an autophagy-related receptor [123], but the role of ubiquitin remains speculative. Overall, the UPS and ribophagy serve as means for degradation of unneeded ribosomal proteins and assembled complexes, respectively, regulating translation by controlling sheer ribosome abundance.

### 3.3. Ubiquitin in Non-Functional rRNA Decay

Similar to ribosomal proteins, rRNA undergo ubiquitin-dependent processes of degradation that contribute to translational control. Mutations or damage in rRNA that remove functionality in either the decoding center or the peptidyl transferase center of the ribosome often lead to non-functional rRNA decay (NRD) [124]. Separate NRD pathways occur for both the 18S rRNA in the small subunit and the 25S rRNA in the large subunit [124] (Figure 3). Both mechanisms are mediated by ubiquitin; however, it is unknown how the ubiquitin enzymes involved sense and are recruited to ubiquitinate ribosomes with a nonfunctional rRNA. For 18S NRD, the Inada lab recently overexpressed a mutated, nonfunctional form of 18S rRNA (A1492C) in yeast, and found increased K63 polyubiquitination of the ribosomal protein uS3 at K212 [113] (Figure 2). Additionally, monoubiquitination of uS3 at K212 by the E3 ligase Mag2 was characterized as the first step before extension into K63 polyubiquitin chains [113]. This elongation of K63 linkages can be performed by either of the E3 ligases Hel2 or Rsp5 [113], highlighting a redundancy in this mechanism and implying its importance in cellular function. The K63 ubiquitination itself allows for ribosome dissociation, providing a 40S subunit as a substrate for NRD and promoting 18S rRNA degradation [113]. Parallel studies focusing on 25S NRD in yeast showed that overexpression of a mutated, nonfunctional form of 25S rRNA (A2451U) led to increased ubiquitination of the ribosome. It was also shown that deletion of the E3 ligase Rtt101 led to stabilization of these 25S rRNAs [124,125], suggesting that Rtt101 is responsible for ubiquitination of the 60S ribosome during 25S NRD. Rtt101 was shown to conjugate K48 polyubiquitin chains, as a K48R ubiquitin mutant strain (unable to form K48 chains) prevented the formation of this ubiquitin signal on the ribosome [126]. After Rtt101 ubiquitinates the ribosome, the Cdc48-Npl4-Ufd1 complex binds to the ubiquitin chain and promotes subunit dissociation [126], showing a role for ubiquitin in subunit dissociation that is similar to the 18S NRD. In addition to providing a signal for subunit dissociation, K48 polyubiquitination sends ribosomal proteins of the inactive ribosome to the UPS, which allows for RNases to access the 25S rRNA [126]. Interestingly, 25S NRD but not 18S NRD is dependent on UPS function, as proteasomal inhibition by MG-132 prevents decay of nonfunctional 25S rRNA and not 18S rRNA [126]. The ubiquitination of the ribosome provides many intricate roles for specific rRNA degradation pathways, from allowing access to the rRNA itself to recruiting factors for dissociation and subsequent clearance of defective molecules.

## 4. Role of Ubiquitin in Ribosome-Associated Protein Quality Control

The Ribosome-associated protein Quality Control (RQC) pathway is a protective cellular process by which ribosomes that become stalled during translation are recognized, split, and recycled. This ubiquitin-mediated pathway is essential for rescuing ribosome machinery, degrading faulty mRNA transcripts, and eliminating incompletely translated polypeptides to avoid aggregation and cell toxicity [128,129]. In this pathway, ubiquitin is involved in two different steps: the resolution of stalled ribosomes and the degradation of the arrested polypeptide (Figure 3). In both steps, ubiquitin acts as a signaling molecule, either recruiting factors for ribosome dissociation, or recruiting shuttling factors that will drive these peptides to the proteasome for degradation. Additionally, various forms of ubiquitin linkages have been reported to be involved in the RQC, highlighting the broad impact ubiquitin has on this pathway. Although ubiquitin is involved along the progression of this pathway, initiation of the RQC is induced by the stalling of an actively translating ribosome. Stalling occurs through a variety of causes such as aa-tRNA insufficiency or mRNA truncation, with the most studied example being ribosome stalling at a 3′ UTR polyadenylated (poly(A)) tail due to the absence or readthrough of a stop codon [129,130]. In addition to translation through the poly(A) tail, a truncation from endonucleolytic cleavage can also causing stalling at the 3′ end. It has been shown that the stalling of the ribosome at the 3′ end is sensed by the protein PELO (Dom34 in yeast), and stalled ribosomes that are collided at the poly(A) tail are recognized by the E3 ligase ZNF598 (Hel2 in yeast) [51,131,132,133], leading to the ubiquitination of 40S ribosomal proteins. A number of RQC factors and nucleases are subsequently recruited to recognize and split the ribosome by its subunits and degrade the mRNA [134]. While the 40S subunit is recycled, the nascent and arrested polypeptide chain in the 60S subunit undergoes ubiquitination by the E3 ligase Listerin (Ltn1 in yeast), leading to its degradation [135,136,137]. The embedded tRNA is removed, and the 60S subunit is recycled along with the remaining RQC components. This section will further explore these two ubiquitination steps within the RQC pathway and highlight their impact on quality control and translational regulation.

### 4.1. Ubiquitination of the Stalled Ribosome

The first ubiquitination event of a stalled ribosome by ZNF598 serves as a key regulatory event for RQC progression. ZNF598 is a 98 kDa E3 ligase that ubiquitinates the small subunit ribosomal proteins uS10 and eS10 upon ribosome collisions [131,132] (Figure 2). ZNF598 requires the E2 ligase UBE2D3 (Ubc4/5 in yeast) to modify the small subunit of the ribosome, and has a RING-type finger domain near the N-terminus that provides the scaffold between the E2 and the ribosomal proteins [51,138]. It has been shown that a mutation at its active residue in the RING domain (C29A) blocks its ubiquitinating activity [131], and a knockdown of ZNF598 causes readthroughs of poly(A) sequences that usually would cause stalling [51]. Although the linkage types involved in this process have yet to be unambiguously confirmed, studies in yeast have suggested that Hel2 function depends on K63 polyubiquitination [64]. Hel2 has also been reported to facilitate the K63 polyubiquitination of eS7 after monoubiquitination by the E3 ligase Not4, further supporting its role as an E3 ligase conjugating K63 linkages [139]. However, overexpression of a K48R ubiquitin mutant (unable to form K48 chains) in yeast has been shown to reduce the ubiquitination of uS10 at K6 and K8 [51]. The same study proposed that monoubiquitination or diubiquitination by K48 linkages occurs at uS10 [51], broadening the discussion of the linkage types involved in the RQC pathway. A recent study demonstrated that ZNF598 recruitment and subsequent eS10 ubiquitination occurs specifically during a low-dose, incomplete translation inhibition by emetine that induces random stalling and collisions [132]. Structural data show that a scaffolding protein called RACK1 stabilizes the interface of the ribosome collision. Although the full molecular detail of the interaction between the ribosomal proteins, ZNF598, and RACK1, is currently being explored, the leading hypothesis suggests that RACK1 brings together the target sites of ZNF598 ubiquitination on uS3 and uS10 during ribosome collision [131,139]. The knockdown of RACK1 also leads to poly(A) readthroughs, further supporting its hypothesized role in the ubiquitination event [131,139,140]. However, the order of recruitment of these enzymes to the small subunit has yet to be elucidated. Additionally, further downstream RQC events are shown to be facilitated by this 40S ubiquitination. One of these pathways, named no-go mRNA decay (NGD), targets the mRNA within stalled ribosomes for degradation, starting with endonucleolytic cleavage [141]. Importantly, both the NGD and RQC pathways are initiated by translation arrest, and are dependent on RACK1 [140,142,143]. Ribosome collisions were proposed to trigger NGD-induced mRNA cleavage [144], with the di-ribosome (disome) being the minimal requirement and the minimal ribosome collision unit to couple NGD and RQC through Hel2 [139]. An endonuclease called Cue2, which has four UBDs, has been shown to be recruited to ribosomes following ubiquitination of eS10 by Hel2 in yeast, to cleave faulty mRNA during collisions [134]. This work supports ubiquitin’s role as a direct signal to recruit downstream effectors of NGD. The discovery of DUBs that antagonize the 40S ubiquitination would provide an interesting mechanism to counteract the RQC pathway. DUBs in the RQC pathway have begun to be explored, as a recent study showed that among 58 DUBs screened, overexpression of either USP21 or OTUD3 in mammalian cells can cause the removal of ZNF598-specific ribosomal ubiquitination [145]. Thus, the ubiquitination of the stalled ribosome serves as a regulatory point for not only the RQC, but multiple other quality control pathways as well.

### 4.2. Ubiquitination of the RQC Arrested Peptides

The second ubiquitination event in the RQC pathway is involved in the degradation of the arrested polypeptide chain protecting the cell from aggregation-prone products. After recognition and splitting, the 60S-peptidyl-tRNA complex is met by a number of components of the RQC complex. One of these components, a scaffolding protein called NEMF (Rqc2/Tae2 in yeast), binds to the subunit interface of the 60S subunit and is used as a docking site for the large, 200 kDa E3 ligase Listerin (Ltn1 in yeast) [135,146]. The association between Listerin and the 60S subunit only occurs in the presence of NEMF, as recent in vitro studies demonstrated that NEMF is necessary and sufficient for Listerin stabilization on the 60S subunit [146]. Due to its elongated structure, Listerin is able to simultaneously bind by its N-terminal domain to the ‘p-stalk’ of the 60S subunit, and by its C-terminal domain to the solvent-exposed surface by the tunnel exit site [147]. A T61A mutation at the N-terminal domain of Listerin has been demonstrated to disrupt its interactions with both NEMF and the ‘p-stalk’ of the 60S [147]. Once the complex is formed, Listerin adds a polyubiquitin chain to the incomplete polypeptide [148].

As do most E3 ligases, Listerin conjugates ubiquitin onto lysine residues on its substrate; however, nascent chains might not always have exposed lysine residues outside of the 60S subunit for ubiquitination to occur. In the case that lysine residues are inaccessible to Listerin while buried in the exit tunnel, cells utilize the C-terminal addition of alanine and threonine (a process called CATylation), which enhances ubiquitination by Listerin [149,150]. A study demonstrated that nascent chains become stabilized in an CATylation-incompetent mutant [149], emphasizing the importance of CATylation for degradation of the peptide. In *LTN1-*deleted cells, these CAT tails become stabilized in an Rqc2-dependent manner [149]. Interestingly, the CAT tail can also serve as a degradation signal itself when Listerin fails to ubiquitinate the polypeptide [149]. The combination of ubiquitination and CATylation provides multiple manners by which cells can promote proteasomal degradation of nascent and arrested polypeptide chains.

In the next stage of the RQC pathway, the ubiquitin on the arrested chain facilitates its degradation by shuttling the peptide to the proteasome. In this process, the AAA+ ATPase VCP (Cdc48 in yeast) binds to the ubiquitin signal and shuttles the peptides for degradation through the proteasome. VCP has been shown to facilitate the degradation of various ubiquitinated proteins in many other contexts through its ubiquitin binding domain [151]. Studies in yeast have shown accumulation of ubiquitinated nascent peptides (that are linked to tRNAs) in mutants of *CDC48* [137]. Much is still unknown about the ubiquitination of the nascent chain, such as whether an E2 conjugase participates in this interaction, and the mechanism as to how VCP removes the ubiquitinated arrested peptide from the ribosome. On the organismal level, perturbations in the process of nascent chain ubiquitination is known to lead to neurodegenerative phenotypes, likely related to protein aggregation [152]. Specifically, mouse with mutations in Listerin were shown to have quicker degeneration of motor and sensory root axons, typically seen in amyotrophic lateral sclerosis (ALS) [153]. Further characterization will be vital to the understanding of Listerin and ubiquitin role in cell physiology and health. Although much of the understanding of the RQC pathway and the role of ubiquitin has emerged within the last decade, the knowledge is central in the current and future comprehension of translational control.

## 5. Role of Ubiquitin in Oxidative Stress Response

One important role of ubiquitin in translational regulation has been shown during cellular response to stress. When exposed to stressful conditions, such as oxidative stress, heat shock, proteotoxic stress, and UV radiation, cells must adapt to internal and external factors that pose a risk to survival [154,155,156,157]. Oxidative stress is a prominent stress condition that occurs when the amount of reactive oxidative species (ROS) within the cell overwhelms its antioxidant capacity [158]. ROS are important signaling molecules but can also damage cellular infrastructure due to their reactive nature [158]. During oxidative stress, eukaryotic cells accumulate large amounts of ubiquitinated proteins [65,159], and much of the earlier research was focused on the degradation role of ubiquitin in the removal of oxidatively damaged proteins [159,160,161,162,163]. Although protein degradation is an important mechanism to control ribosome abundance (discussed in Section 3), more recent work has focused on the non-degradative role ubiquitin plays in translational regulation under oxidative stress [50,164,165]. Regulation of gene expression at the translational level is a process that allows for quick alteration of global protein levels for adequate adaptation [166]. As ROS are such a threat, there are several levels of protein production control, including global inhibition of translation coupled to synthesis of antioxidant proteins [167]. A recent study showed that ubiquitination acts as part of the unfolded protein response (UPR) and contributes to a decrease in protein synthesis during ER stress [156]. Treatment with UV was also found to induce site-specific ubiquitination events on the ribosome, suggesting a role for ribosomal ubiquitination in multiple stress pathways [156]. Although ubiquitin seems to play an important role in global stress response, here we will focus on an oxidative-stress specific pathway in which translation is regulated by K63 polyubiquitination of ribosomal proteins. This ubiquitin-mediated translational control is an essential mechanism for maintaining cell health during the oxidative stress response.

### 5.1. Translational Control under Oxidative Stress

Regulation of protein synthesis is essential for response to oxidative stress due to its damaging nature. Synthesis of new proteins is often halted or decreased due to risks associated with high ROS, including protein damage, toxic gain-of-function, and aggregation [168]. ROS-driven modification of ribosomal proteins and rRNA alters the function and efficiency of the ribosomes, as extensively reviewed by Shcherbik and Pestov [165]. These modifications include the K48 polyubiquitination of ribosomal proteins, which leads to their degradation by the canonical UPS [163]. In addition to causing damage, ROS can also act as a signaling molecule, interacting with regulatory enzymes, thereby activating adaptive cellular responses [169]. Thus, regulation of translation occurs by altering binding interactions of ribosomal proteins with translation factors, mRNA, and tRNA [170]. One of the most well-studied pathways of translational control during oxidative stress is the regulation of initiation mediated by phosphorylation of eIF2α [171]. ROS result in the phosphorylation of eukaryotic initiation factor-2α (eIF2α) via activation of the yeast amino acid control kinase *GCN2*, by a still uncharacterized mechanism [164]. In humans, four distinct kinases can perform the phosphorylation of eIF2α [172]. Phosphorylated eIF2α has an increased affinity to eIF2B, thus reducing the availability of ternary complex necessary for initiation, and global inhibition of translation [173]. This inhibition is lost upon deletion of *GCN2,* resulting in active initiation during oxidative stress [164]. However, many mechanisms of translation control post-initiation have been proposed [50,164,174]. Unlike the global inhibition of general protein synthesis, cells must upregulate the expression of antioxidant proteins to cope with oxidative stress. Reduction in the ternary complex levels increases translation of the transcription factor Gcn4 due to bypassing of regulatory upstream ORFs (uORFs) [175]. Expression of Gcn4 induces transcription of genes encoding proteins that will cope with oxidative stress including various antioxidant enzymes and general stress defense proteins [167,170,176,177]. The upregulation of detoxifying proteins is also seen at the level of transcription, with an increase in transcript amounts seen with exposure to both superoxide and hydrogen peroxide [178]. Despite the thorough control of translation by Gcn2, a limited reversion of translation in a *gcn2Δ* mutant suggests the presence of a *GCN2*-independent level of control beyond initiation [164].

### 5.2. Overview of Redox Control of Translation by Ubiquitin (RTU) Pathway

The Redox control of Translation by Ubiquitin (RTU) pathway has been proposed as a regulatory mechanism to rapidly control translating ribosomes during oxidative stress. Works from the Silva lab have shown that K63 polyubiquitin heavily accumulates on ribosomes during cellular exposure to hydrogen peroxide and rapidly declines during recovery from stress [65]. This rapid accumulation depends on redox reactions and K63 ubiquitin is able to modify both active and inactive monosomes, and polysomes [50]. Rad6 and Bre1 were identified as the E2-E3 pair involved in the K63 assembly on the ribosome [65], a new role for the complex, which has been previously studied for its role in histone ubiquitination [179]. In a screen of E2s, only the deletion of *RAD6* decreased the accumulation of K63 ubiquitin upon oxidative stress [65]. Rad6 coupled to different E3s has shown to be involved in several pathways, including DNA repair (Rad6-Rad18) [180], histone modification (Rad6-Bre1) [179], and protein degradation (Rad6-Ubr1) [181]. Out of the aforementioned E3s, only the *bre1Δ* strain showed a loss of K63 accumulation during oxidative stress [65]. Bre1 is a RING E3, which interacts with its substrate and its dedicated E2 [182]. The structural details on how Bre1 and Rad6 interact with the ribosome and among each other remains to be understood; however, in other pathways, the arginine motif within the Bre1 RING domain orients Rad6 to modify histone lysine 123 of histone H2B [183]. These motifs and residues might also play important structural roles in the RTU allowing the recruitment and stabilization of these ubiquitin enzymes on the surface of the ribosome to promote K63 polyubiquitination.

To identify ribosomal targets of K63 ubiquitin under stress, a new mass spectrometry technique was developed that quantified ubiquitin sites with linkage specificity [50]. When proteins are digested with trypsin protease, as is often used in mass spectrometry, a digylcyl motif (GG) from the last two amino acids of ubiquitin’s C-terminus leaves a distinct remnant (K-ε-GG) that has been used to determine the common sites of ubiquitination on target proteins [184,185]. Challenges in identifying linkage-specific sites have arisen due to the separation of the ubiquitin chain and its substrate during trypsin digestion. Therefore, previous methods either identified sites of ubiquitin though GG modification [41,42,43,185] or quantified the abundance of individual ubiquitin chains by determining the levels of their signature peptides [105,186,187]. This new technique involves a sequential enrichment that initially targets K63 polyubiquitin and following trypsin digestion, enriches for target residues via the GG remnant. Using this new technique, the Silva lab identified 78 K63 ubiquitinated sites on 37 ribosomal proteins during oxidative stress [50]. Forty-five of these sites, including the most abundant, are concentrated on the head of the small subunit (Figure 2). The head of the 40S is an important region of translational regulation due to the binding of translational regulators such as initiation and elongation factors, as well as mRNA and tRNA molecules [188]. The collection of ubiquitin sites on the solvent-exposed surface may impact binding of these factors as a mean of translational control. It was shown that K63 ubiquitin acts independently of initiation inhibition and that resumption of initiation is insufficient to restore normal translational, suggesting a role in post-initiation regulation [65]. K63 ubiquitin has also been shown to stabilize polysomes within the cell, as a K63R strain, in which the 63rd lysine of ubiquitin is replaced with an arginine to prevent K63 polyubiquitin formation, showed a decrease in polysomes and an increase in unassembled subunits [65,189]. Collectively, these data support the idea that K63 ubiquitin is an important regulator of translation under stress, likely regulating protein synthesis at the elongation stage.

Because ribosomal K63 ubiquitin accumulates, is readily reversed, and does not lead to proteasomal degradation during the recovery phase of stress [65], the Silva lab investigated the role of deubiquitinating enzymes in cellular recovery. In mammalian cells, members of the ubiquitin-specific protease family, a subfamily of cysteine protease DUBs, were found to be reversibly inhibited under oxidative stress [190,191,192]. Screening for potential DUBs in yeast showed that deletion of *UBP2* leads to the accumulation of K63 ubiquitin [65]. It was also showed that the DUB activity of Ubp2 is inhibited by hydrogen peroxide in vitro and that inhibition could be reversed with reduction by DTT [65]. It has been shown that Ubp2 prefers to target K63 polyubiquitin and is known to antagonize other K63 building ubiquitin ligases such as Rsp5 [193]. Silva and colleagues thus showed that Ubp2-mediated hydrolysis of K63 ubiquitin chains can be redox regulated [65]. It was proposed that upon inactivation of Ubp2 by ROS, Rad6 and Bre1 ubiquitinate the ribosome, resulting in an accumulation of K63 linked chains that regulates translational elongation in repressing protein synthesis (Figure 3). This novel mechanism of translational control found in both yeast and mammals presents a unique role for ubiquitin and a redox-sensitive DUB in the regulation of protein synthesis.

Although H_2_O_2_ is required to induce RTU by regulating the activity of selective K63 ubiquitin enzymes [65], we are only in the beginning of understanding the complexity of translation regulation during stress. As mentioned before, oxidative stress can impact translation at several steps of the process, including initiation, elongation, and quality control [50,164,165]. Recent work by the Zaher lab showed that RNA alkylating and oxidizing agents, MMS and 4-NQO, respectively, can lead to ribosome stalling and increased Ltn1-dependent ubiquitination of the arrested peptide, in addition to ubiquitination of ribosomal proteins mediated by the RQC E3 Hel2 [194]. Therefore, RTU and RQC pathways seem to be important to control specific subpopulations of ribosomes that might arise depending on the intensity of the stress, its duration, and the chemical nature of the reactive oxygen species employed. Understanding the interplay between RTU and RQC might provide further ways to control protein abundance, express antioxidant proteins, and remove damaged components during oxidative stress. Oxidative stress poses a risk to cell viability and has been implicated in several diseases, including neurodegenerative diseases, cardiovascular disease, and aging [195,196,197]. The control of translational output during occurrences of oxidative stress is an important mechanism for minimizing damage inflicted by ROS. A better understanding of these defensive mechanisms will aid in addressing the physiological problems associated with stress-related diseases. At the technological level, several barriers remain to fully understand how ubiquitin regulates translation.

## 6. Concluding Remarks

As our understanding of the global role of ubiquitination expands from a degradation signal to a modulator of a vast array of cellular pathways, a deeper review of ribosome ubiquitination gives insight into the multiple roles ubiquitin can play within translation control. In the past ten years, several regulatory mechanisms of translational control by ubiquitination of ribosomes have been explored, including the control of ribosome abundance [49], quality control at the ribosome [128], and the translational response to oxidative stress [65,165]. Within these mechanisms, we have discussed how ubiquitin participates in changing ribosomal structure, impacting interactions with ribosome binding factors, and signaling to degrade various components of the ribosome. These mechanisms are all vital to global cellular function, as tight regulation of translation is necessary for proper gene expression and effective use of cellular energy.

As mentioned previously, recent literature has highlighted functional heterogeneity among subpopulations of ribosomes [28]. The functional role of the subpopulations of ubiquitinated ribosomes has yet to be elucidated and could provide evidence for translational reprogramming through the targeting of transcripts, recognition of specific factors, or even through subcellular localization. Even though many of these ubiquitin-mediated translational control pathways have been identified, many questions still remain about how the ubiquitin chains, enzymes involved, and subcellular locations of these events contribute to this functional heterogeneity. The growing complexity of the ubiquitin chain in linkage, length, and further PTMs provides several mechanistic modes on how ubiquitin can further control translation. Post-translational modifications on ubiquitin itself such as phosphorylation and acetylation have been reported [73,74] but are understudied and could potentially impact stability and recognition of ubiquitin signals on the individual ribosomes. Various linkage types such as mono, K63, and K48 ubiquitin have reported roles in translational control, but how linkage type confers specificity to the manner ubiquitin is recognized on the ribosome remains unknown as well. In addition, whether the chain length of the ubiquitin signal can be modulated by additional ubiquitin ligases and DUBs also remains to be elucidated [198]. Current mass spectrometry methodologies utilize the GG remnant to identify ubiquitin modified sites, but this method falls short in identifying chain length and the increasing complexity of linkage types, modifications, and dynamics. Recent techniques, including UbiSite, present improvements in the identification of bona fide ubiquitin specific peptides and can also capture linear M1 ubiquitin targets by using an antibody that recognizes the 13 most C-terminal residues of ubiquitin, respectively, instead of GG-lysine [43,50]. Despite these advances, a way to directly determine site-specific ubiquitin linkages at the systems level has yet to be developed.

Ribosome subpopulations are also defined by the set of enzymes associated with the ribosome [28], such as E2, E3, and DUBs. Further exploration in how ubiquitin enzymes interact and provide specific linkage types onto their ribosomal protein substrates will help to understand how these ribosomal subpopulations are regulated and how much redundancy and complexity this system can present. A deeper understanding of the regulation of the ubiquitin enzymes themselves regarding their expression and post-translational regulation would provide knowledge on the prevalence and function of each of these subpopulations. Finally, another layer of regulation can be defined by the subcellular location of ribosome particles and each ubiquitin enzyme involved [28,199]. A comprehensive characterization of their expression profile and subcellular localization would provide answers to how ubiquitin enzymes compete for binding surfaces and substrates. Subcellular localization could impact ribosomal binding kinetics upon different physiological conditions and determine their physiological role. The recent technological advancements in next-generation sequencing, structural biology, and mass spectrometry highlighted throughout this work can help us answer many of these questions and advance the field of translation control mediated by ubiquitin.

## Figures and Tables

**Figure 1 ijms-21-01151-f001:**
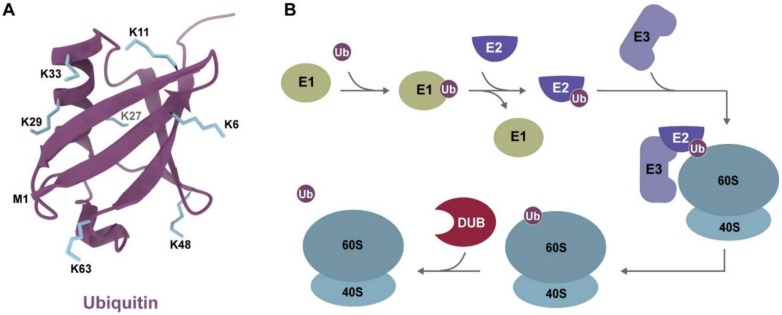
Ubiquitin and overview of the process of ubiquitination. (**A**) Structure of ubiquitin and the position of its lysine residues. The variety of roles for ubiquitin is in part mediated by the seven lysine residues (K) and the amino group of the first methionine residue (M1), where ubiquitin can form further polyubiquitin linkages. (**B**) Summary of the ubiquitin enzymatic cascade, in which a target, such as ribosomal proteins, can be ubiquitinated by the sequential functions of E1 activating enzyme, E2 conjugating enzymes, and E3 ligases. Ubiquitin modifications can lead to signaling functions or be reversed by deubiquitinating enzymes (DUB) [80,81,82].

**Figure 2 ijms-21-01151-f002:**
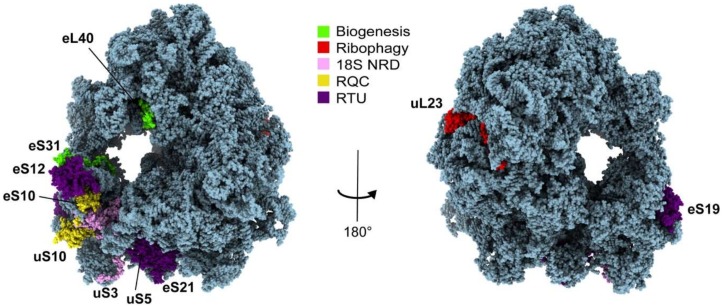
Spatial organization of ubiquitin-modified ribosomal proteins in translational control. Ribosomal proteins eL40 and eS31 are involved in ribosome biogenesis (green), uL23 is involved in ribophagy (red), uS3 is involved in 18S nonfunctional rRNA decay (pink), and eS10 and uS10 are involved in ribosome quality control (yellow). The ribosomal proteins found to be highly K63 ubiquitinated during oxidative stress and possibly involved in the RTU pathway (uS5, eS12, eS19, eS21) are highlighted in purple. This list includes uS3 and uS10, which are also involved in 18S NRD and RQC, respectively [81,82,127].

**Figure 3 ijms-21-01151-f003:**
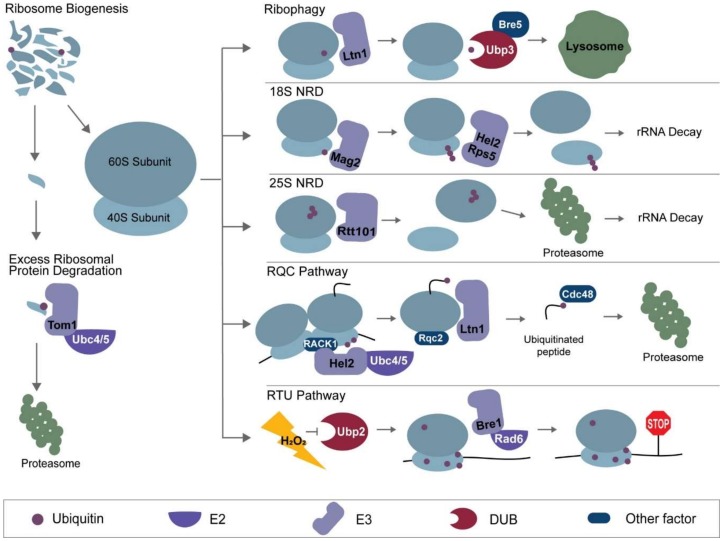
Summary of the ubiquitin-mediated pathways of translational control. The left panel highlights processes of ribosome turnover (ribosome biogenesis and excess ribosomal protein degradation). The right panel highlights ribosome fates through ubiquitin-mediated mechanisms, namely ribophagy, 18S non-functional rRNA decay (NRD), 25S NRD, Ribosome-associated protein Quality Control (RQC), and Redox control of Translation by Ubiquitin (RTU). Proteins involved in each mechanism are labeled as their yeast homologs, although some of these pathways have also been explored in mammalian systems.

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
