# Peer review of "Expanding Role of Ubiquitin in Translational Control"

_ijms, 2020, doi:10.3390/ijms21031151_

Round 1
Reviewer 1 Report
This review by Dougherty et al does an excellent job at describing and discussion recent advances in the field on the role of Ubiquitin (Ub) in translational control in eukaryotes. The manuscript is laid out well, has very good graphics to help guide the reader towards easier conceptualization, and identifies key issues moving forward. The amount of depth provided in each section is adequate, informative and thought-provoking. Overall, an excellent review that will find a wide audience not just in the Ub field, but also beyond.
Author Response
Review #1
This review by Dougherty et al does an excellent job at describing and discussion recent advances in the field on the role of Ubiquitin (Ub) in translational control in eukaryotes. The manuscript is laid out well, has very good graphics to help guide the reader towards easier conceptualization, and identifies key issues moving forward. The amount of depth provided in each section is adequate, informative and thought-provoking. Overall, an excellent review that will find a wide audience not just in the Ub field, but also beyond.
We appreciate the reviewer’s comments and the support for our manuscript. Our goal was indeed to reach a broader community in ubiquitination, translational control, and cell biology and we are glad to hear that the reviewer agrees with it.
Reviewer 2 Report
This review compiles the known data on the role of the Ubiquitin Proteasome System (UPS) in translational control. This topic is highly important and may be interesting also to scientists not directly working on translation.
The review is well written and includes a wide and appropriate bibliography and the figures are helpful. I just have minor suggestions.
Page 3: Among the role of K63 chains, the driving of proteins to lysosomes (autophagy) is missing. This is an important link between the UPS and autophagy Page 5 – paragraph 2.3: The following article (de la Peña et al., Science 362, 2018) seems more appropriate than ref 102 for illustrating the role of the AAA ATPases of the 19S regulatory complex for substrate translocation. Page 5 – paragraph 3: The authors wrote that cellular stress, nutrient availability and “other causes” modify ribosome abundance. What “other causes”? As no reference is provided, the reader has to guess. Please specify. Page 11, 1st paragraph, line 6: “… global inhibition coupled to synthesis of …” Figure 3 legend: Please add that NRD stands for “non-functional rRNA decay”
Author Response
Review #2
This review compiles the known data on the role of the Ubiquitin Proteasome System (UPS) in translational control. This topic is highly important and may be interesting also to scientists not directly working on translation.
The review is well written and includes a wide and appropriate bibliography and the figures are helpful. I just have minor suggestions.
We are pleased to hear that the reviewer sees the topic addressed in this manuscript as important and relevant to a wide audience and we are pleased with the positive reviews. Please see a point-by-point answer to your suggestions.
Page 3: Among the role of K63 chains, the driving of proteins to lysosomes (autophagy) is missing. This is an important link between the UPS and autophagy
We added an opening statement that mentions K63 ubiquitin’s role in autophagy and how autophagy could affect ribosomes in a non-specific fashion. We then further address ribophagy in more comprehensive manner.
Page 5 – paragraph 2.3: The following article (de la Peña et al., Science 362, 2018) seems more appropriate than ref 102 for illustrating the role of the AAA ATPases of the 19S regulatory complex for substrate translocation.
We appreciate the suggestion and added de la Peña’s article to our reference list.
Page 5 – paragraph 3: The authors wrote that cellular stress, nutrient availability and “other causes” modify ribosome abundance. What “other causes”? As no reference is provided, the reader has to guess. Please specify.
Thank you for the pointer: This sentence now reads as “Perturbations in ribosome abundance can occur during environmental changes including cellular response to stresses [106] and limited nutrient availability [107]”. The respective references were added accordingly.
Page 11, 1st paragraph, line 6: “… global inhibition coupled to synthesis of …”
Thank you noticing this typo. It has been fixed in the current version of the manuscript.
Figure 3 legend: Please add that NRD stands for “non-functional rRNA decay”
Thank you for the pointer. NRD abbreviation is now present in the legend of figure 3.
Reviewer 3 Report
The review "Expanding Role of Ubiquitin in Translational Control" by Dougherty et al. delivers an excelent overview on the current research on multiple ubiquitin-mediated mechanisms of translational control. The review is well written and very informative.
Author Response
Review #3
The review "Expanding Role of Ubiquitin in Translational Control" by Dougherty et al. delivers an excellent overview on the current research on multiple ubiquitin-mediated mechanisms of translational control. The review is well written and very informative.
We want to thank and appreciate the reviewer for this very positive evaluation of our work.